# LEARNING STRUCTURED COMMUNICATION FOR MULTI-AGENT REINFORCEMENT LEARNING

## ABSTRACT

Learning to cooperate is crucial for many practical large-scale multi-agent applications. In this work, we consider an important collaborative task, in which agents learn to efficiently communicate with each other under a multi-agent reinforcement learning (MARL) setting. Despite the fact that there has been a number of existing works along this line, achieving global cooperation at scale is still challenging. In particular, most of the existing algorithms suffer from issues such as scalability and high communication complexity, in the sense that when the agent population is large, it can be difficult to extract effective information for high-performance MARL. In contrast, the proposed algorithmic framework, termed Learning Structured Communication (LSC), is not only scalable but also learns efficiently. The key idea is to allow the agents to dynamically learn a hierarchical communication structure, while under such a structure the graph neural network (GNN) is used to efficiently extract useful information to be exchanged between the neighboring agents. A number of new techniques are proposed to tightly integrate the communication structure learning, GNN optimization and MARL tasks. Extensive experiments are performed to demonstrate that, the proposed LSC framework enjoys high communication efficiency, scalability, and global cooperation capability.

## 1 INTRODUCTION

Reinforcement learning (RL) has achieved remarkable success in solving single-agent sequential decision problems under interactive and complicated environments, such as games (Mnih et al., 2015; Silver et al., 2016) and robotics (Lillicrap et al., 2016). In many real world applications such as intelligent transportation systems (Adler & Blue, 2002) and unmanned systems(Semsar-Kazerooni & Khorasani, 2009), not only one, but usually a large number of agents are involved in the learning tasks. Such a setting naturally leads to the popular multi-agent reinforcement learning (MARL) problems, where the key research challenges include how to design scalable and efficient learning schemes under an unstationary environment (caused by partial observation and/or the dynamics of other agents' policies), with large and/or dynamic problem dimension, and complicated and uncertain relationship between agents.

Learning to communicate among agents has been regarded as an effective manner to strengthen the inter-agent collaboration and ultimately improve the quality of policies learned by MARL. Various communication-based MARL algorithms have been devised recently, e.g., DIAL (Foerster et al., 2016), CommNet (Sukhbaatar et al., 2016), ATOC (Jiang & Lu, 2018), IC3Net (Singh et al., 2019) and TarMAC (Das et al., 2019). These schemes aim to improve the inter-agent collaboration by learning communication strategy to exchange information between agents. However, there are still two bottlenecks unresolved, especially when faced a large number of agents.

One bottleneck lies in that achieving effective communication and global collaboration is difficult with limited resources, such as narrow communication bandwidth and energy. In particular, DIAL and TarMAC require each agent to communicate with all the other agents, i.e., a fully-connected communication network (Figure 1(a)), which is not feasible for large scale scenarios with geographically apart agents. CommNet and IC3 assume a star network (Figure 1(b)) with a central node coordinating the global collaboration of agents, which again does not allow large scale scenarios with long range communications. ATOC introduces an interesting attention scheme to build a tree

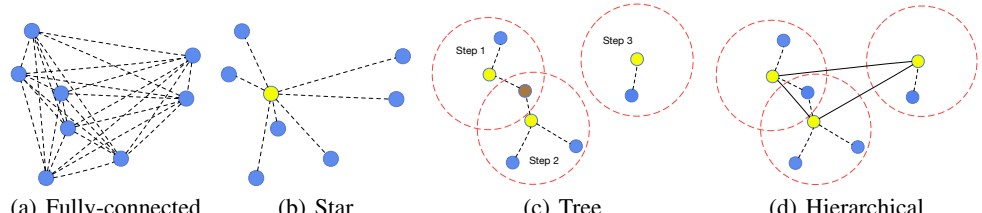

Figure 1: Topology of different communication structures and LSC falls into the hierarchical one.

communication network (Figure 1(c)). While the tree network can be scaled, global collaboration has to be realized through inefficient multi-hop and sequential communications. In a word, improper communication topologies will limit the cooperation ability in large scale scenarios.

Another bottleneck is the difficulty of extracting essential information to exchange between agents for achieving high-performance MARL, especially when the number of agents grows. Most of the existing works simply concatenate, take the mean or use the LSTM to extract information to be exchanged. First two lack in considering the inter-relationship between agents, and LSTM assumes that there is a fixed sequence of message passing between agents, that is, the relationship between agents is predefined. Recently, TarMAC utilized an attention scheme to aggregate messages by considering the relationship from each agent to all others. However, the improper communication topology still hinders the information extraction. The communication structure needs to be jointly designed with the information extraction scheme to achieve further improved learning performance.

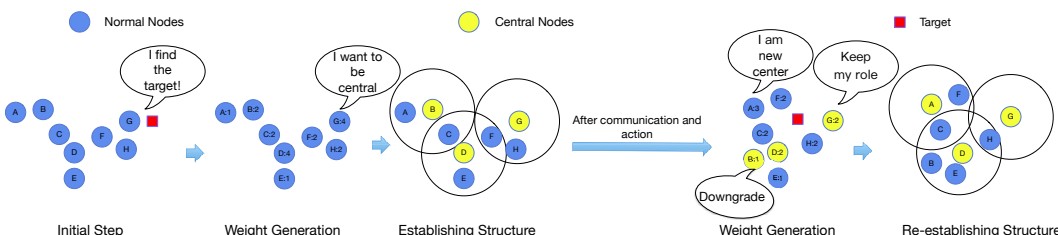

Figure 2: Procedure for dynamically establishing structured communication network. **Left**: Each agent determines its communication importance weight based on partial local observation. For instance, the agent "G" finds the target (red square), then it will be possible to get a higher weight "4" and become the central. **Right**: The importance weight generation step and network construction step will be repeated iteratively. After communication and action procedures, agents will generate their new communication importance weights, and determine to keep or change their roles respectively. Further, the structured communication network will be re-established.

To address the above two issues, we propose a novel structured communication-based algorithm, called *learning structured communication* (**LSC**). Our LSC combines a structured communication network module and a communication-based policy module, which aims to establish a scalable hierarchically structured network and information exchange scheme for large scale MARL. In particular, a hierarchically structured communication network (Figure 1(d)) is dynamically learned based on local partial observations of agents. In the hierarchically structured network, all agents are grouped into clusters, where global collaboration can be achieved via intra-group and inter-group communications. In contrast to the other three types in Figure 1, the proposed hierarchical communication network is more flexible and scalable, with fewer resources needed to achieve long-range and global collaboration.

The procedure to establish such a hierarchically structured communication network is shown in Figure 2. To better utilize the relationship between agents given the hierarchically structured communication network and obtain more effective information extraction, graph neural network (GNN) (Scarselli et al., 2008) is employed. In GNN, each communication step involves information embedding and aggregation. Benefiting from the unordered aggregation power and the dynamic graph adaptability of GNN, the proposed LSC algorithm can extract valuable information effectively. The GNN-based information extraction procedure is depicted in Figure 3.

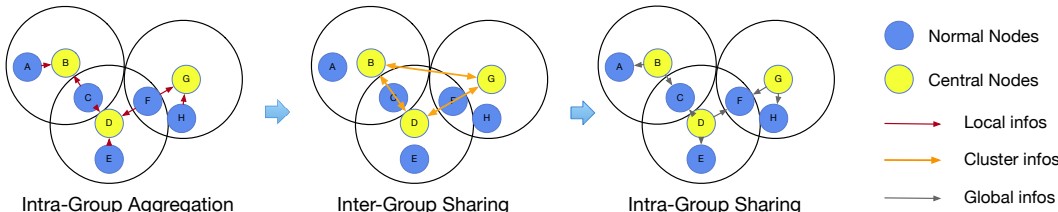

Figure 3: GNN-based communication extraction procedure. Each node denotes an agent. The edge embedding can be considered as the communication message. The network learning procedure properly fits the communication procedure, and effectively learn valuable messages involving the global network structure and agents relationship. **Left**: Low-level normal agents transfer their local valuable embeddings to the associated central agents. **Middle**: High-level central agents communicate with each other to gain a sense of global perception. **Right**: All central agents broadcast embedding information to their normal agents to form global cooperation.

This paper is devoted to the learning of communication structure among agents. To our knowledge, this is the first work of hierarchical structured learning to communication for MARL. It allows to learn communication structure adaptively instead of using predefined forms. Specifically:

i) To improve scalability for a large number of agents, a hierarchical structure is devised that divides the agents into higher-level central agents and sub-level normal ones. As such, the communication network is sparsified. While it still allows for more effective global cooperation via message passing among the central agents, compared with the star/tree structures.

ii) For effective communication and global cooperation, the message representation learning is deeply integrated into the information aggregating and permeating through the network, via graph neural network (GNN), which is a natural combination with the hierarchical communication structure.

iii) Extensive experiments on both MAgent and StarCraft2 show our approach achieves state-of-the-art scalability and effectiveness on large-scale MARL problems.

## 2 RELATED WORK AND PRELIMINARIES

Many multi-agent reinforcement learning algorithms without communication in the inference procedure have experienced fast development. Recent works like MADDPG (Lowe et al., 2017), QMIX (Rashid et al., 2018), COMA (Foerster et al., 2018) and MAAC (Iqbal & Sha, 2019) adopt a centralized training and decentralized implementing framework. All agents' local observations and actions are considered to improve the learning stability. These algorithms are generally not suitable for large-scale case due to explosive growing number of agents.

Communication-based MARL algorithms have been showed effective for large-scale agent cooperation. Earlier works assume that all agents need to communicate with each other. DIAL (Foerster et al., 2016) learns to communication through back-propagating all other agents' gradients to the message generator network. Similarly, CommNet (Sukhbaatar et al., 2016) sends all agents' hidden states to the shared communication channel and further learns the message based on the average of all other hidden states. MFRL (Yang et al., 2018) approximates the influence of other agents by averaging the actions of surrounding neighbor agents, which could mitigate the dimensional disaster for large-scale cases. However, this can be considered as a predefined communication pattern, which is unable to adapt to complex large-scale scenarios. Communication between all agents will lead to high communication complexity and difficulty of useful information extraction. DGN (Jiang et al., 2018) employs graph convolution network (GCN) to extract relationships between agents which could result in better collaboration. However, it considers all agents equivalently and assumes the communication of each agent has to involve all neighbor agents which limits to adapt to more practical bandwidth-limited environments. IC3Net (Singh et al., 2019) uses a communication gate to decide whether to communicate with the center, but adopt the same star structure like CommNet which requires high bandwidth and can hard to extract valuable information with only one center. ATOC (Jiang & Lu, 2018) and TarMAC (Das et al., 2019) introduce the attention mechanism to determine when to communicate and whom to communicate with, respectively. TarMAC focuses

more on message aggregation rather than the communication structure. SchedNet (Kim et al., 2019) aims to learn a weight-based scheduler to determine the communication sequence and priority. From the perspective of employing GNN into MARL, MAGNet (Malysheva et al., 2018) that utilizes a relevance graph representation of the environment and a message passing mechanism to help agents learning. However, it requires heuristic rules to establish the graph which is hard to achieve in complex environments. RFM (Tacchetti et al., 2019) use graph to represent the relationship between different entities, aiming to provide interpretable representations.

Before the main method, we introduce some preliminaries to facilitate the presentation.

**Partial Observable Stochastic Games.** In stochastic games, agents learn policies by maximizing their cumulative rewards through interacting with the environment and other agents. The partial observable stochastic games (POSG) can be characterized as a tuple $\langle \mathcal{I}, \mathcal{S}, b^0, \mathcal{A}, \mathcal{O}, \mathcal{P}, \mathcal{P}_e, \mathcal{R} \rangle$ where $\mathcal{I}$ denotes the set of agents indexed from 1 to $n$; $\mathcal{S}$ denotes the finite set of states; $b^0$ represents the initial state distribution and $\mathcal{A}$ denotes the set of joint actions. $A_i$ is the action space of agent $i$, $\mathbf{a} = \langle a_1, \cdot, a_n \rangle$ denotes a joint action; $\mathcal{O}$ denotes the joint observations and $O_i$ is the observation space for agent $i$, $\mathbf{o} = \langle o_1, \cdot, o_n \rangle$ denotes a joint observation; $\mathcal{P}$ denotes the Markovian transition distribution with $P\left(\tilde{s}, \mathbf{o} \middle| s, \mathbf{a}\right)$ as the probability of state $s$ transit to $\tilde{s}$ and result $\mathbf{o}$ after taking action $\mathbf{a}$. $\mathcal{P}_e(\mathbf{o}|s)$ is the Markovian observation emission probability function. $\mathcal{R} : \mathcal{S} \times \mathcal{A} \rightarrow \mathbb{R}^n$ means the reward function for agents. The overall task of the MARL problem can be solved by properly objective function modeling, which also indicates the relationship among agents, e.g., cooperation, competition or mixed.

**Graph Neural Network.** Graph neural network (GNN) (Scarselli et al., 2008) is a deep embedding framework to handle graph-based data on a graph $\mathcal{G} = (\mathcal{V}, \mathcal{E})$. $\mathbf{v}_i$ denotes the node feature vector for node $v_i \in \mathcal{V}$ (for $N_v$ nodes), $\mathbf{e}_k$ denotes the edge feature vector for edge $e_k \in \mathcal{E}$ (for $N_e$ edges) with $r_k, s_k$ be the receiver and sender of edge $e_k$ respectively. The vector $\mathbf{u}$ denotes the global feature. The graph network framework in (Battaglia et al., 2018) is employed, which divides computation on graph data to several blocks to gain flexible processing ability. Each block introduces the aggregation and embedding functions to handle graph data. There are many variants of GNN, like message-passing neural network (Gilmer et al., 2017) and non local neural networks (Wang et al., 2018). By treating every agent as a node and each communication message exchanging as the edge in a graph, the observations and messages as the attributes of nodes and edges, respectively. The whole communication process can be formulated to a graph neural network. The relationships among agents can be effectively extracted to enable efficient communication message learning.

**Independent Deep Q-Learning.** Deep $Q$-Network (DQN) (Mnih et al., 2015) is popular in deep reinforcement learning, which is one of the few RL algorithms applicable for large-scale MARL. In each step, each agent observes state $s$ and takes an action $a$ based on policy $\pi$. It receives reward $r$ and next state $\tilde{s}$ from environment. To maximize the cumulative reward $R = \sum_t r_t$, DQN learns the action-value function $Q^\pi(s, a) = \mathbb{E}_{s \sim \mathcal{P}, a \sim \pi(s)} [R_t | s_t = s, a_t = a]$ by minimizing $\mathcal{L}(\theta) = \mathbb{E}_{s,a,r,\tilde{s}} [\tilde{y} - Q(s, a; \theta)]$, where $\tilde{y} = r + \gamma \max_{\tilde{a}} Q(\tilde{s}, \tilde{a}; \theta)$. The agent follows $\epsilon$-greedy policy, that is, selects the action that maximizes the $Q$-value with probability 1-$\epsilon$ or randomly. The Independent Deep $Q$-Learning (IDQN) (Tampuu et al., 2017) is an extension of DQN by ignoring the influence of other agents for multi-agent case. Every agent learns a $Q$-function $Q^a(u^a|s; \theta^a)$ based on its own observation and received reward.

Our algorithm employs DQN as the basic RL algorithm based on the following two considerations: 1) our algorithm is dedicated to discuss the learning communication mechanism in large-scaleMARL scenarios, as a result we can choose a concise and effective basic RL algorithm like thewell-known DQN; 2) data collection in large-scale MARL environments is extremely inefficiently,while DQN has excellent data efficiency as an offline RL algorithm.

## 3  LSC: LEARNING STRUCTURED COMMUNICATION

Our communication architecture has two key modules: *structured communication network module* and *communication-based policy module*, shown in Figure 4. The first module aims to establish the dynamic hierarchical structured communication network in a distributed fashion, while the second module contains the GNN-based communication extraction and $Q$-network components. Without loss of generality, we use DQN as the basic reinforcement algorithm, however our approach can incorporate any value-based or actor-critic methods. The details of LSC is depicted in Algorithm 1

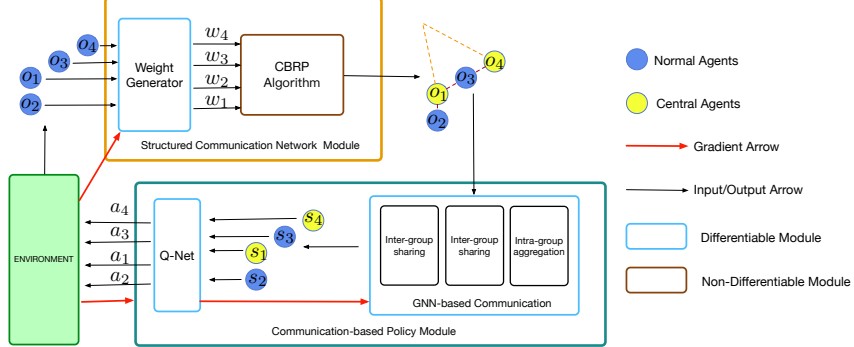

Figure 4: Algorithm framework of LSC with *Structured Communication Network Module* and *Communication-based Policy Module*, where $s_i$, $o_i$, $a_i$ and $w_i$ denote state (global perception), observation, action and importance weight of agent $i$. The former module uses partial observation to establish the communication structure. The latter employs GNN-based communication and $Q$-Network to extract communication content and produces collaboration policies respectively based on established communication structure.

---

**Algorithm 1 LSC**: Learning Structured Communication

---

1: Initialization: weight generator parameters $\theta^w$, $Q$-net parameters $\theta^Q$, GNN parameters $\theta^{gnn}$, target $Q$-net parameters $\theta^{\tilde{Q}}$, replay buffer $\mathcal{R} = \varnothing$, group radius $d$, the number of agents $n$;
2: **for** Episode $= 1, \cdots, M$ **do**
3:      Reset $t = 0$, global state $s^t$ and observation $o_i^t$ for each agent $i$, Normal agents set $\mathcal{V}_n^t = \{$all agents$\}$ and $\mathcal{V}_c^t = \varnothing$;
4:      **for** $t = 1, \cdots, T$ and $s_t \neq$ terminal **do**
5:          **for** each agent $i$ **do**
6:              With probability $\epsilon$ pick a random action $w_i^t$ else $w_i^t = \arg\max_{\{w_i\}} Q_{\theta^w}(o_i^t)$;
7:          Get current position $\text{POSs}_i^t$ of each agent $i$;
8:          $(\mathcal{V}_n^t, \mathcal{V}_c^t, \mathcal{E}) = \text{CBRP}((\mathcal{V}_n^{t-1}, \mathcal{V}_c^{t-1}), \{w_1^t, \cdots, w_n^t\}, \{\text{POSs}_1^t, \cdots, \text{POSs}_n^t\}, d)$;
9:          $\{q_1^t, \cdots, q_n^t\} = \text{HCOMM}(\mathcal{V}_n^t, \mathcal{V}_c^t, \mathcal{E})$;
10:          **for** each agent $i$ **do**
11:              With probability $\epsilon$ pick a random action $a_i^t$ else choose the action that has the largest
12:              value in the vector $q_i^t$;
13:          Execute global actions and get global reward $r^t$, next state $s^{t+1}$, next observation $o^{t+1}$;
14:          Get updated position $\text{POSs}_i^{t+1}$ for each agent $i$;
15:          Store $(s^t, o^t, \{\text{POSs}_1^t, \cdots, \text{POSs}_n^t\}, a^t, r^t, o^{t+1}, \{\text{POSs}_1^{t+1}, \cdots, \text{POSs}_n^{t+1}\}, s^{t+1})$ to $\mathcal{R}$;
16:      **for** $k = 1, \cdots, K$ **do**
17:          Sample a random mini-batch transitions from $\mathcal{R}$;
18:          Update weight generator $\theta^w$ by minimizing Eq. (1);
19:          Update communication based policy module $(\theta^Q, \theta^{gnn})$ by minimizing Eq. (2);
20:          Update the target networks through Eq. (3).

---

Specifically, the CBRP function automatically and distributively establishes the structured communication network based on the learnt importance weights. The HCOMM function denotes the communication-based policy module, which outputs the $Q$-values based on the GNN-based communication messages. Both CBRP and HCOMM are discussed in the following subsections, and the details of CBRP and HCOMM can be found in Appendix.

## 3.1 STRUCTURED COMMUNICATION NETWORK MODULE

The structured communication network module takes the role of establishing a hierarchical structured communication network which will be employed in communication-based policy module. Two sub-modules are included, i.e., the weight generator and the Cluster Based Routing Protocol (CBRP). The weight generator sub-module aims to determine the importance weight for each agent automatically. It is modeled through a neural network $f_{wg} : o \rightarrow w$, where the weight $w$ can mea-

sure the confidence of an agent to become a center. Further, the CBRP sub-module employs the weights of all agents $\{w_i\}$ to construct the hierarchical structured communication network. To emphasize, the CBRP sub-module can be implemented in a distributed fashion, as a result, the central agents can be elected distributedly. This advantage ensures the practicability for large-scale case.

The CBRP method (Rezaee & Yaghmaee, 2009) is a typical method for establishing a hierarchical routing structure. The key idea for CBRP is that each agent will check whether central agent or agent that has larger weight $w$ exists in its receptive area. The agent will become a central agent if no above agent is found, else it will keep its own role. With enough checking steps, each agent will either be an central agent or in some central agents' receptive. All agents can be separated into several groups with each central agent as the group leader. The overall hierarchical structured communication network further will be established by fully connecting all central agents from different groups and connecting the agents in each group to their central agent.

There is a strong connection between these two modules in LSC algorithm. Different generated weights will lead to different hierarchical structured communication network, which would cause diverse performance of the communication-based policy. Some experiment results also have confirmed that the weights have a great influence on the performance, which motivates us to train these two modules end-to-end. However the CBRP sub-module is not differentiable, which means the gradients cannot be back-propagated from communication-based policy module to the weight generator sub-module. Therefore, we introduce another RL task as an auxiliary, i.e., each agent takes its weight as an action by treating the communication-based policy module as an *extra* unobservable part of the environment, and receiving the same reward as the main RL task in the communication-based policy module discussed below. Moreover the weight $w$ is constrained in the integer set $\{0, 1, 2, 3, 4\}$. The action space becomes discrete, as a result DQN algorithm can be used again to train the weight generator. At this time, the weight generator can be regarded as a $Q$-value function. The loss $\ell(\theta^w)$ for the weight generator sub-module becomes clear as follows, with $y_i = r_i + \gamma \max_{\tilde{w}_i} Q_{\theta^w}(\tilde{o}_i, \tilde{w}_i)$. $r_i$ denotes the reward received for agent $i$ from environment.

$$\ell(\theta^w) = \mathbb{E}_{\mathbf{o}, \mathbf{w}, r, \tilde{\mathbf{o}}} \Big[ \sum_{i=1} (Q_{\theta^w}(o_i, w_i) - y_i)^2 \Big]. \tag{1}$$

## 3.2 COMMUNICATION-BASED POLICY MODULE

After the structured communication network topology is determined, the communication-based policy module will learn the communication content and generate the final global collaboration policy. The communication-based policy module consists of two sub-modules, i.e., GNN-based communication sub-module and the $Q$-Net sub-module. The first one aims to learn the communication messages and further update overall state perception, while the other sub-module learns the policy based on the new state perceptions after efficient communication. Different from many existing works (Foerster et al., 2016; Das et al., 2019; Singh et al., 2019), the agents play differently in the GNN-base communication sub-module. Central agents should guarantee high-level information and dominate the agents in their driven groups respectively. The hope is that such a structure can ensure the effectiveness of communication and the efficiency of intra-group and inter-group collaboration.

Recall Figure 3, the well-established hierarchical structured communication network can be represented by a tuple $(\mathcal{V}, \mathcal{E})$ while the edges are directed. The node set $\mathcal{V}$ contains $N_v$ nodes which can be divided into the central node set $\mathcal{V}_c$ and the normal node set $\mathcal{V}_n$. For central node $i \in \mathcal{V}_c$, the node feature vector $v_i$ includes the embedding feature $v_i^n$, the central role feature $v_i^c$ and the global feature $v_i^g$; for normal node $i \in \mathcal{V}_n$, the node feature vector $v_i$ only includes the embedding feature $v_i^n$. For each edge $(i \rightarrow j) \in \mathcal{E}$ with $i, j \in \mathcal{V}$, the edge feature vector is denoted as $e_{ij}$. Functions $\phi$ and $\rho$ denote the update embedding function and aggregate function respectively. As shown in Figure 3, the overall GNN-based communication sub-module consists of three steps, and the GNN operation is detailed in Table 1 and as follows:

**Step 1: Intra-group aggregation.** In each group, the normal agent embeds their local information and sends it to the associated central agent $j \in \mathcal{V}_c$; the central agent aggregates the information from all associated normal agents and updates its central role feature;

**Step 2: Inter-group sharing.** The central agent communicates with the other central agent with cluster information, further aggregates the received and indicates the global perception;

**Step 3: Intra-group sharing.** The central agent communicates all its feature with the associated normal agents while the normal agent aggregates the received information from central agents. Both the embedding feature of central and normal agents will be updated.

Table 1: GNN-based Communication Architecture

| Type | Edge $(i \to j) \in \mathcal{E}$ | Edge Update Scheme | Node Update Scheme |
|---|---|---|---|
| Step 1: intra-group aggregation | $i \in \mathcal{V}_n, j \in \mathcal{V}_c$ | $e_{ij} = \phi(v_i^n), \bar{e}_j = \rho(\{e_{ij}\}_{(i \to j) \in \mathcal{E}})$ | $v_j^c = \phi(\bar{e}_j, v_j^n)$ |
| Step 2: inter-group sharing | $i \in \mathcal{V}_c, j \in \mathcal{V}_c$ | $e_{ij} = \phi(v_i^c, v_i^n), \bar{e}_j = \rho(\{e_{ij}\}_{(i \to j) \in \mathcal{E}})$ | $v_j^g = \phi(\bar{e}_j, v_j^n)$ |
| Step 3: intra-group sharing | $i \in \mathcal{V}_c, j \in \mathcal{V}_n \cup \mathcal{V}_c$ | $e_{ij} = \phi(v_i^g, v_i^c, v_i^n), \bar{e}_j = \rho(\{e_{ij}\}_{(i \to j) \in \mathcal{E}})$ | $v_i^n = \phi(\bar{e}_i, v_i^n), v_j^n = \phi(\bar{e}_j, v_j^n)$ |

Table 2: Comparison of different MARL algorithms for communication efficiency.

| Algorithm | DIAL | CommNet | IC3 | ATOC | LSC |
|---|---|---|---|---|---|
| $N_{msg}$ | $\mathcal{O}(n^2)$ | $\mathcal{O}(n)$ | $\mathcal{O}(n)$ | $\mathcal{O}(kb)$ | $\mathcal{O}(k^2 + kb)$ |
| $N_{step}$ | $\mathcal{O}(1)$ | $\mathcal{O}(1)$ | $\mathcal{O}(1)$ | $\mathcal{O}(d)$ | $\mathcal{O}(1)$ |
| $N_{b\text{-}r}$ | $\mathcal{O}(n)$ | $\mathcal{O}(n)$ | $\mathcal{O}(n)$ | $\mathcal{O}(b)$ | $\mathcal{O}(\max(b, k))$ |

The GNN-based communication sub-module is modeled as a GNN ($f_{\theta^{gnn}}$) with parameter $\theta^{gnn}$, while the following $Q$-Net of agent $i$ ($Q_{\theta^Q}^i$) is parameterized by shared parameter $\theta^Q$. The gradient can be back-propagated from $Q$-Net to the graph neural network, as a result the overall loss of communication based policy module is as follows:

$$\ell(\theta^Q, \theta^{gnn}) := \mathbb{E}_{\mathbf{o}, \mathbf{a}, r, \tilde{\mathbf{o}}} \left[ \sum_{i=1}^n \left( Q_{\theta^Q}^i(f_{\theta^{gnn}}(\mathbf{o}), a_i) - y_i \right)^2 \right], \quad (2)$$

where $y_i = r_i + \gamma \max_{\tilde{a}_i} Q_{\theta^Q}^i(f_{\theta^{gnn}}(\tilde{\mathbf{o}}), \tilde{a}_i)$. $r_i$ denotes the reward received for agent $i$ from environment. Some softly updating scheme is further employed to update target network, i.e.,

$$\theta^{\tilde{Q}} = \tau \theta^Q + (1 - \tau)\theta^{\tilde{Q}}, \quad \text{and} \quad \theta^{g\tilde{n}n} = \tau \theta^{gnn} + (1 - \tau)\theta^{g\tilde{n}n}. \quad (3)$$

Here we discuss communication efficiency from three aspects: the number of message exchanging ($N_{msg}$) among agents; the number of steps during the communication procedure ($N_{step}$); the communication bandwidth and range requirements for each agent ($N_{b\text{-}r}$), and $n$ is the total number of agents. The details about communication efficiency are presented in Table 2 for each stage of the MARL algorithms. For DIAL, each agent communicates with all other agents based on the fully-connected network, which results in $\mathcal{O}(n^2)$ message exchanging complexity. DIAL need only one communication step, however the communication bandwidth and range requirement for each agent is high and in the order of $\mathcal{O}(n)$. Different from DIAL, CommNet and IC3 both employ the star communication network, as a result the number of message exchanging is in the order of $\mathcal{O}(n)$. $N_{step}$ and $N_{b\text{-}r}$ are the same as DIAL. The communication complexity of ATOC and our proposed LSC depends on the number of groups (denoted as $k < n$, which is automatically determined in the algorithms) and the maximum output degree of the communication network (denoted as $b < n$). For ATOC, the communication network it tree-type, so that it only need to exchange $\mathcal{O}(kb)$ messages. However, the number of steps is larger for ATOC for its sequential property and is in the order of $\mathcal{O}(d)$ ($d$ denotes the depth of the communication network). $N_{b\text{-}r}$ will become much smaller to be $\mathcal{O}(b)$ because communication happens in groups. Furthermore for our LSC, the number of message exchanging is a bit larger than ATOC due to the communication among all elected centers, i.e., $\mathcal{O}(k^2 + kb)$. However the depth of the hierarchical communication network is only two which results in $\mathcal{O}(1)$ communication steps, while $N_{b\text{-}r}$ is in the order of $\mathcal{O}(\max(b, k))$. Overall, our LSC algorithm has advantage in the communication efficiency.

## 4 EXPERIMENTS

We compare LSC with state-of-the-art MARL methods in two large-scale battle environments, i.e., the grid world platform MAgent (Zheng et al., 2017) and StarCraft2 (Samvelyan et al., 2019), to evaluate their performances from aspects of both network structure and communication.

### 4.1 LARGE SCALE BATTLE GAME IN MAGENT

**Settings.** In a MAgent battle, agents fighting against enemies in a $40 \times 40$ grid world. Each agent only receives its local observation, acts independently and cooperatively, and further gains its reward. The goal for each agent is to attack its enemies and prevent them from being attacked. Each agent

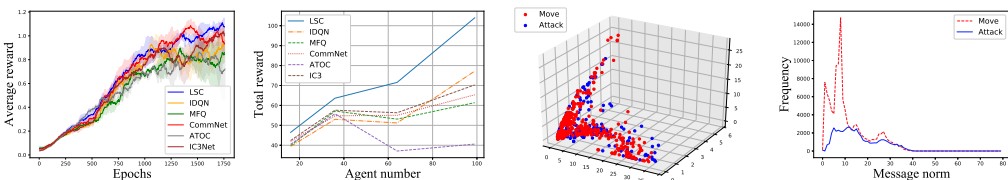

(a) Average reward  (b) Total testing reward  (c) Message distribution  (d) Message norm
Figure 5: Reward of algorithms and message pattern visualization in the MAgent environment.

from both sides has a $6 \times 6$ visual field and can attack its 8 adjacent grids. The speed, attack power and health point for each agent are 1, 1 and 4, which are increased to 2, 2 and 10 for the enemy to increase the difficulty. The reward is $+5$ for successfully attacking an enemy, $-2$ for being killed and $-0.01$ for attacking a blank grid.

**Baselines.** To evaluate the effects of communication scheme, three peer methods on learning to communicate for MARL, i.e., CommNet, IC3 and ATOC, are chosen to compete with our proposed LSC. Considering the weakness of mean aggregation, we replace the aggregation function of CommNet and IC3 with GNN which is same as LSC. The group radius is 6, the same as the visual field, in ATOC and LSC. All communication messages are embedded to 3-dimension vectors for cost-effectiveness. Besides, two MARL methods with no communication, i.e., IDQN and MFQ are also compared, since they are widely used in large-scale environments. In MAgent, the policy of enemy is pretrained by IDQN.

**Policy performance.** Figure 5(a) and Table 3 show the overall performances of compared algorithm in a 64 vs. 64 battle. The learning curves in Figure 5(a) present the average reward of different agents by epoch. LSC achieves better rewards quickly after 700 epochs and finally converges to an obvious higher point (about 1.15) than baselines. Table 3 and Table 4 give quantitative comparisons of these methods. Each algorithm is given 50 trials with its well-trained model. *Mean-reward*, $N_{kill}$, $N_{dead}$, $Ratio_{kd}$ in Table 3 denote the mean of average final rewards of agents, the number of killed enemies and dead agents, and the ratio $N_{kill}/N_{dead}$, respectively. Following Figure 5(a), LSC can obtain a better mean reward stably, with a 30% performance advantage at least. $N_{kill}$ are similar, because all approaches fulfill the mission, and beat the pre-trained IDQN. It is achieved by LSC with the least casualty, i.e., the smallest $N_{dead}$ and the highest $Ratio_{kd}$. Table 4 gives the comparisons in terms of the number of epochs to achieve the same reward value from 0.7 to 1.2 within maximal 1750 epochs in the training procedure. One can see that LSC needs fewer epochs to achieve the same reward compared with all the other algorithms, while more reward can be guaranteed within the maximal epochs. These results indicate that LSC can promote collaboration and cooperation, and produce superior policies.

**Communication effectiveness.** As observed from the blue and orange curves in Figure 5(a) and the first and fifth columns in Table 3, that LSC outperforms IDQN greatly. Since IDQN is the special case of LSC without communication procedure, this phenomenon demonstrates the usefulness of our proposed communication solution. Meanwhile, it can be noted that LSC also surpasses other MARL with communication algorithms, which is the consequence of its advanced structure. As mentioned in our experiments, CommNet and IC3 adopt the same message dimension and the same aggregation function as LSC, which leads to better performances than the original versions. However, the star structure makes the center node need to process all agents' information in CommNet and IC3. When the agent number increases, the message extraction could be difficult. Thus, the final performances cannot be compared with the LSC. For ATOC in large scale environments, the message needs to jump multiple times between local circles, and multiple information aggregation and extraction bring in approximation error, which results in policy deterioration.

**Scalability.** Figure 5(b) shows the total reward curves of team by the agent number (10-99). Especially when the team has 99 agents, the reward of LSC is 1.45-2.75 times of other methods. It can be seen that the structured communication of LSC confers superior performances at different scales, because it utilizes the divide-and-conquer strategy to automatically group local agents and aggregate centers. In Figure 5(b) When the agent number is less than 80, the communication between agents can help to learn policy, so the orange dashed line is almost bellow other lines. When the number is more than 80, the demand of star-style information processing exceed the ability of aggregation

Table 3: Performance comparisons in terms of average mean-reward, numbers and ratio of kills and death (64 vs. 64 agents, in 50 testing trials). The bold stands for the best result in each row.

|  | LSC | CommNet | IC3 | ATOC | IDQN | MFQ |
|---|---|---|---|---|---|---|
| $Mean\text{-}reward$ | **1.11** | 0.86 | 0.93 | 0.58 | 0.8 | 0.83 |
| $N_{kill}$ | **62.6** | 61.3 | 62.48 | 61.46 | 62.1 | 62 |
| $N_{dead}$ | **28.9** | 31.64 | 32.0 | 51.42 | 32.3 | 31.4 |
| $Ratio_{kd}$ | **2.16** | 1.93 | 1.95 | 1.20 | 1.92 | 1.97 |

Table 4: Comparisons on the used epoch number to achieve same reward of the training procedure in the MAgent environment.

| Reward | LSC | CommNet | IC3 | ATOC | IDQN | MFQ |
|---|---|---|---|---|---|---|
| 0.7 | **769** | 1199 | 787 | 933 | 1133 | 1254 |
| 0.8 | **828** | 1292 | 935 | 1151 | 1460 | 1304 |
| 0.9 | **1051** | 1525 | 1508 | 1471 | – | 1426 |
| 1.0 | **1271** | – | 1588 | 1506 | – | – |
| 1.1 | **1413** | – | 1619 | – | – | – |
| 1.2 | **1704** | – | – | – | – | – |

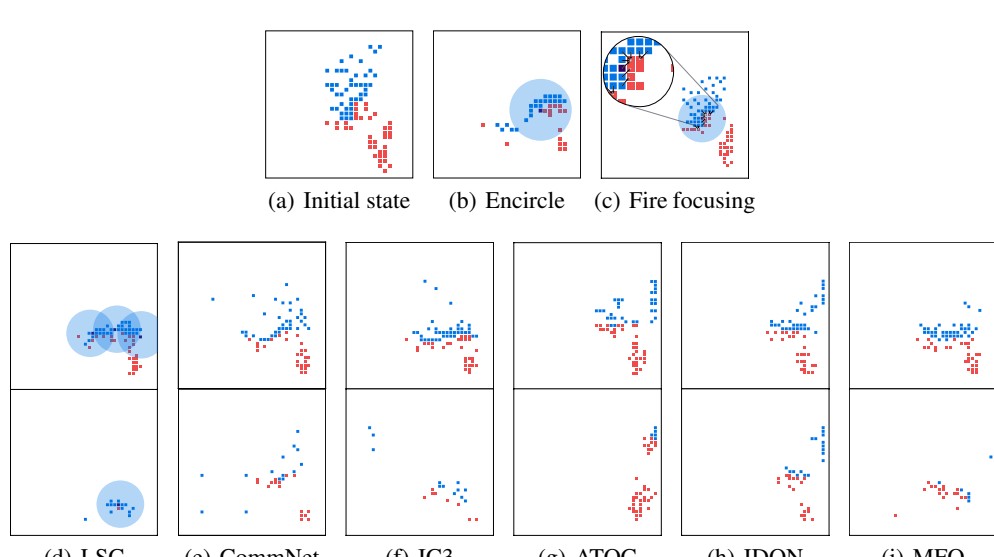

(a) Initial state    (b) Encircle    (c) Fire focusing

(d) LSC    (e) CommNet    (f) IC3    (g) ATOC    (h) IDQN    (i) MFQ

Figure 6: Behavior illustration. The first row shows two typical behavior by LSC. In the second row, the top and bottom plot denote the early state and the near to final battle state, respectively.

network, thus CommNet and IC3 are inferior to IDQN. For ATOC, the jump of message becomes the bottleneck as the agent number increases, as inspected in the analysis above. To sum up, the experiments indicate LSC obviously has better scalability than the baseline.

**Message visualization.** To analyze the communication messages learned by GNN, we execute LSC with its well-trained policy 50 times, and visualize the 3 dimensional vector sent by central nodes in Figure 5. Observing Figure 5(c) and 5(d), after receiving messages, the majority of agents choose to *move*, and the minority choose to *attack*. This means that agents in LSC is very positive to adjust the team formation and then cooperate to attack. It is worthy to notice that most messages have a small norm (less than 10), and the norm of a large proportion is around 0. From the aspect of optimization, the redundant messages with larger norm will bring in more noise to other agents. In this way, LSC minimizes the impact of redundant messages on the final performance. This similar phenomenon that central nodes is nearly silent in many cases, is also mentioned by CommNet (Sukhbaatar et al., 2016). Therefore, the message representation module via the GNN-based communication module can generate meaningful and efficient messages theoretically and empirically.

**Behavior pattern.** Here, we demonstrate the battle tactics evolved for MARL via structured communication. To analyzed the globally cooperation strategy of LSC, we visualize the progress of the

battle. For fairness, we start six algorithms from the same state in Figure 6(a). The typical behavior patterns learnt by LSC are presented in the figures of the first row. Via the intra-group communication and cooperation, in Figure 6(b), the blue team (LSC) organizes an encirclement to nearby red enemies; the team has a local numerical superiority, and focus agents' fire to wipe out enemies, denoted by the black attack arrows in Figure 6(c). These show the intra-group cooperation of LSC. Upper figures in the second row of Figure 6 shows the situation of early stage (17 steps) after initialization for six methods, and lower figures show the states after 50 steps. From these results, we may arrive a conclusion that the team with our LSC can beat the opponent more quickly with a more aggressive policy. By intra-group and inter-group collaboration, Figure 6(d) LSC has carried out encircling and fire-focusing many times, and achieves an enormous advantage within only 17 steps, and wipe out the enemies within 50 steps. For both IDQN and MFQ, agents tend to cooperate within their visual range, and once the agents get separated out of visual range, they can hardly form global cooperation, which lead to the failure result in Figure 6(h) and 6(i). Similarly, agents controlled by ATOC communicate only among group range, thus they encounter similar situations. Agents for CommNet and IC3 have global communication, however once some agents get far away from the central agents, central agents can hardly understand their messages, making ineffective cooperation. As Figure 6(e) and 6(f), some agents get far away from the majority of agents, thus their results are not ideal. To sum up, in Figure 6(d), agents controlled by LSC form a global encircle strategy by communication in both intra-group and inter-group, thus LSC-based agents can wipe out enemies faster than the baselines.

## 4.2 LARGE SCALE BATTLE GAME IN STARCRAFT2

Battle game in StarCraft2 is a confrontation between two marine teams shown in appendix, which is much more complex than MAgent. Specifically, we use a 25 vs. 25 map, i.e., **25m**, where the range vision is 9 and the map size is $1920 \times 1200$. To evaluate the cooperation, agents one team are controlled by the individual learned policy. Here, we compare our LSC with IQDN, Commnet and IC3. The action space consists of movement to an adjacent grid and shooting with the range 6. We adopt the same dense reward setting ($0.44$ for killing an enemy, $8.9$ for winning the battle) as SMAC (Samvelyan et al., 2019). Agents of the enemy are controlled by the built-in game AI.

In Figure 7, LSC outperforms compared algorithms, where the blue reward curve is much higher than others. Although the agent number (25) is fewer than MAgent, the observation and action space of StarCraft2 are much larger, leading to the difficulty to learn policy. Therefore, the performance of IDQN and MFQ degrades notably in this complex environment, while LSC, IC3, CommNet and ATOC outperform it. This is because they entail communication to facilitate cooperation. Moreover, LSC outperforms IC3, CommNet and ATOC. This demonstrates that flexible hierarchical communication and expressive GNN-based message extraction make LSC more qualified for complex tasks than CommNet and IC3's star-style communication.

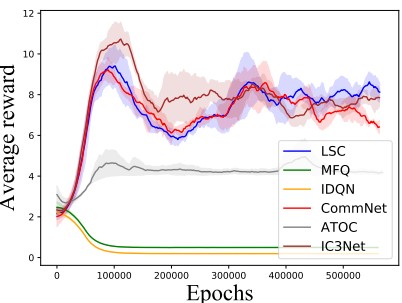

Figure 7: Reward curves on StarCraft2

## 5 CONCLUSION AND FUTURE WORK

In this paper, a novel learning structured communication (LSC) algorithm is proposed for multi-agent reinforcement learning. The hierarchical structure is self-learned by cluster based routing protocol. The communication message representation is naturally embedded and extracted via a graph neural network. Experiments in large-scale games (MAgent and StarCraft2) demonstrated that our LSC can outperform existing learning-to-communicate algorithms with better communication efficiency, cooperation capability, and scalability. In the future, we will improve LSC by considering some practical constraints, such as communication bandwidth and delay.

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

## A  APPENDIX

### A.1  CBRP ALGORITHM AND HCOMM ALGORITHM

---

**Algorithm 2 CBRP**: Cluster Based Routing Protocol

---

1: **function** CBRP($(\mathcal{V}_n^t, \mathcal{V}_c^t), \{w_1^t, \cdots, w_n^t\}, \{\text{POSs}_1^t, \cdots, \text{POSs}_n^t\}, d$)
2:    Define neighbours are distance $< d$, $T_e$ is a constant to control the max-waiting time,$V_u = \varnothing$ is the undecided nodes set and $\mathcal{E} = \varnothing$;
3:    Each node $i$ broadcast its weight $w_i^t$ to neighbours;
4:    **for** $i$ is in central node set $\mathcal{V}_c^t$ **do**                          ▷ Maintain the structure
5:        **if** there is a central nodes in neighbours and its weight is bigger than agent $i$ **then**
6:            Pop node $i$ from $\mathcal{V}_c^t$ and append it to $\mathcal{V}_n^t$;
7:        **for** $i$ is in normal nodes set $\mathcal{V}_n^t$ and no central node is in its neighbour **do**
8:            Pop node $i$ from $\mathcal{V}_n^t$ and append it to $\mathcal{V}_u$;
9:        **for** $i$ is in $\mathcal{V}_u$ concurrently **do**                          ▷ Elect central nodes
10:           **if** does not receive larger weight for $T_e$ **then**
11:               append $i$ to $\mathcal{V}_c^t$ and broadcast to neighbours;
12:           **else**
13:               Wait for the signal from central node for $2T_e$;
14:               **if** received a signal from central node **then**
15:                   append $i$ to $\mathcal{V}_n^t$;
16:               **else**
17:                   append $i$ to $\mathcal{V}_c^t$;
18:       **for** $i$ in $\mathcal{V}_c^t$ **do**                          ▷ generate communication link
19:           **for** $j$ in $\mathcal{V}_i^t$ and $j$ is neighbouring $i$ **do**
20:               append $e_{ij} = 0$ and $e_{ji} = 0$ to $\mathcal{E}$;
21:           **for** $j$ in $\mathcal{V}_c^t$ **do**
22:               append $e_{ij} = 0$ to $\mathcal{E}$;
23:       Return $(\mathcal{V}_n^t, \mathcal{V}_c^t, \mathcal{E})$

---

### A.2  DETAILS AND DISCUSSIONS OF ALL HYPERPARAMETERS

For both MAgent and StarCraft2 environments, the details of all hyperparameters used for approaches are summarized in the following Table 5 and Table 6. All common hyperparameters of all approaches are set to be the same in the same environments. Except for the "ATOC", we change the message dimension to $64$ in both enviroments. For the reason that if the dimension of messages is set to be 3 as other algorithms, the necessary GRU embedding in ATOC leads to failure while $64$ seems to be robust.

For the neural network setting in MAgent enviroment, the node encoder part are all implemented with two convolutional layers with 32 filters and kernel size 3 and a MLP with 256 units. For the $Q$ Encoder part, they are implemented with MLP$(128, 64, 13)$. The message generators for algorithms except for ATOC are implemented with MLP$(64, 32, 3)$ and the aggregation functions are segment sum. While for ATOC, the message generator is MLP$(128, 64)$ and the aggregation functin isGRU$(64)$. As for neural network setting in Starcraft2, we only tune the input and output layer to adapt to the new environment. All these settings can ensure repeatability.

### A.3  BATTLE SCENARIOS OF MAGENT AND STARCRAFT2

### A.4  DISCUSSIONS ON WEIGHT GENERATOR AND GROUP RADIUS

Although LSC utilize the reward feedback to establish the weight generator, which will be various with respect to different settings. To investigate the improvement brought by our learned importance weight generator, we also compare with a basic random weight generator (randomly chosen the central nodes and separate normal nodes into groups). As shown in Figure 9(a), our learned weight generator significantly outperforms randomly. Faster convergence of our learned weight generator

---

**Algorithm 3 HCOMM**: Communication based Policy Module

---

1: **function** HCOMM($\mathcal{V}_n, \mathcal{V}_c, \mathcal{E}$)
2:     ♯ Intra-group aggregation
3:     **for** $v^i$ in $\mathcal{V}_n$ **do**
4:         **for** $v^j$ in $\mathcal{V}_c$ and $(i \rightarrow j)$ in $\mathcal{E}$ **do**
5:             $e_{ij} = \phi^{enc}(v^i);$                       ▷ Generate normal to central messages
6:     **for** $v_j$ in $\mathcal{V}_c$ **do** $\bar{e}_j = \rho(\{e_{ij}\}_{(i \rightarrow j) \in \mathcal{E}});$     ▷ Central agents aggregate received messages
7:         $v_j^c = \phi(\bar{e}_j, v_j^n);$                       ▷ Generate cluster perception
8:     ♯ Inter-group sharing
9:     **for** $v_j$ in $\mathcal{V}_c$ **do**
10:         **for** $v_i$ in $\mathcal{V}_c$ and $(i \rightarrow j)$ in $\mathcal{E}$ **do**
11:             $e_{ij} = \phi(v_i^c, v_i^n);$                  ▷ Generate central to central messages
12:     **for** $v_j$ in $\mathcal{V}_c$ **do**
13:         $\bar{e}_j = \rho(\{e_{ij}\}_{(i \rightarrow j) \in \mathcal{E}});$        ▷ Aggregate received central to central messages
14:         $v_j^g = \phi(\bar{e}_j, v_j^n);$                      ▷ Obtain global perception
15:     ♯ Intro-group sharing
16:     **for** $v_i$ in $\mathcal{V}_c$ **do**
17:         **for** $v_j$ in $\mathcal{V}_n$ and $(i \rightarrow j)$ in $\mathcal{E}$ **do**
18:             $e_{ij} = \phi(v_i^g, v_i^c, v_i^n, e_{ji}), \bar{e}_j = \rho(\{e_{ij}\}_{(i \rightarrow j) \in \mathcal{E}});$     ▷ Generate central to normal messages
19:     **for** $v_j$ in $\mathcal{V}_n \cup \mathcal{V}_c$ **do**
20:         **for** $v_i$ in $\mathcal{V}_c$ and $(i \rightarrow j)$ in $\mathcal{E}$ **do**
21:             $\bar{e}_j = \rho(\{e_{ij}\}_{(i \rightarrow j) \in \mathcal{E}});$        ▷ Aggregate received central to normal messages
22:     $v_j^n = \phi(\bar{e}_j, v_j^n);$                        ▷ Update states
23:     $q_j = Q(v_j^n);$
        **return** $q$.

---

Table 5: Hyperparamaters for Magent

| Parameter | LSC | IDQN | MFQ | CommNet | IC3 | ATOC |
|---|---|---|---|---|---|---|
| Episodes | 1750 | 1750 | 1750 | 1750 | 1750 | 1750 |
| $\epsilon_{start}$ | 1.0 | 1.0 | 1.0 | 1.0 | 1.0 | 1.0 |
| $\epsilon_{end}$ | 0.01 | 0.01 | 0.01 | 0.01 | 0.01 | 0.01 |
| Learning rate | $1e^{-4}$ | $1e^{-4}$ | $1e^{-4}$ | $1e^{-4}$ | $1e^{-4}$ | $1e^{-4}$ |
| Max env steps | 400 | 400 | 400 | 400 | 400 | 400 |
| dimension of messages | 3 | 3 | 3 | 3 | 3 | 64 |
| radius of communication | 6 | $\sim$ | 6 | $\sim$ | $\sim$ | 6 |

Table 6: Hyperparamaters for Starcraft2

| Parameter | LSC | IDQN | MFQ | CommNet | IC3 | ATOC |
|---|---|---|---|---|---|---|
| Total steps | $600k$ | $600k$ | $600k$ | $600k$ | $600k$ | $600k$ |
| $\epsilon_{start}$ | 1.0 | 1.0 | 1.0 | 1.0 | 1.0 | 1.0 |
| $\epsilon_{end}$ | 0.05 | 0.05 | 0.05 | 0.05 | 0.05 | 0.05 |
| Learning rate | $1e^{-4}$ | $1e^{-4}$ | $1e^{-4}$ | $1e^{-4}$ | $1e^{-4}$ | $1e^{-4}$ |
| Max env steps | 10000 | 10000 | 10000 | 10000 | 10000 | 10000 |
| dimension of messages | 3 | 3 | 3 | 3 | 3 | 64 |
| radius of communication | 6 | $\sim$ | 6 | $\sim$ | $\sim$ | 6 |

shows the efficiency brought by involving the reward guided weight generator. The hierarchical structured communication network guaranteed through the learned weight generator improves the communication efficiency. The higher average reward obtained by our learned weight generator also shows the necessity of selecting central nodes based on the learned weight generator rather than randomly chosen.

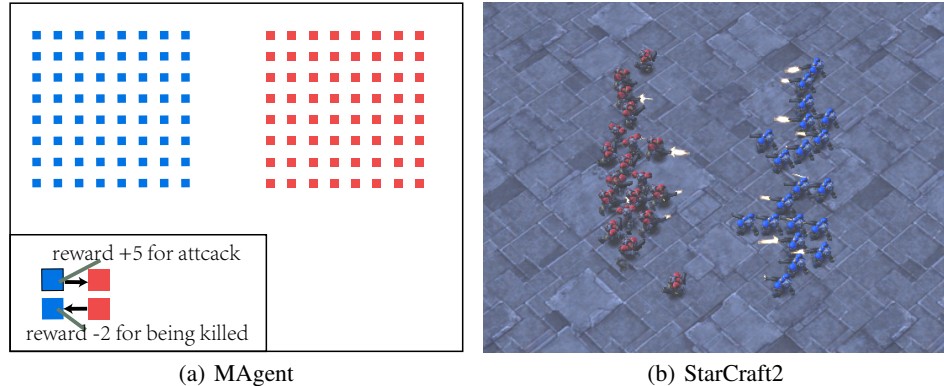

(a) MAgent

(b) StarCraft2

Figure 8: Battle scenarios of MAgent and StarCraft2.

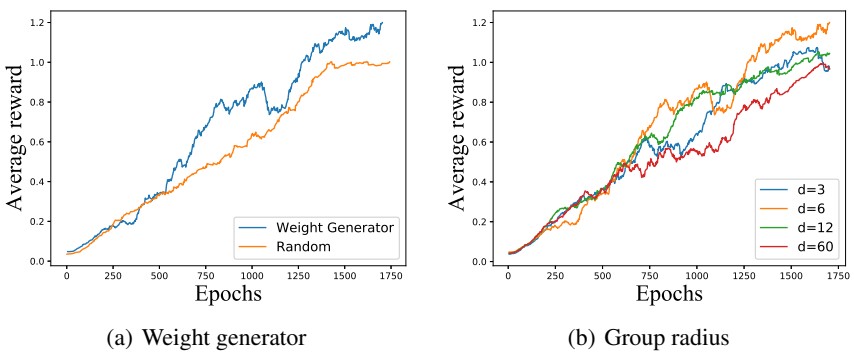

(a) Weight generator

(b) Group radius

Figure 9: Discussions on weight generator and group radius.

To better investigate the hierarchical structure, we compare our LSC algorithm with respect to different group radius $d$, i.e., 3, 6, 12 and 60. As shown in Figure 9(b), LSC with radius 6 outperform other settings. When the radius increasing, the agents can establish inter-groups cooperation easier. However, LSC with radius 60 performs worse than all the other three cases. LSC with extremely large radius will downgrade to CommNet which is not an effective way to extract valuable information.

