# OpenReview forum: "Learning Structured Communication for Multi-agent Reinforcement Learning"
_ICLR.cc/2020/Conference — Reject_

### Official Review · AnonReviewer1 · 2019-10-23
**Official Blind Review #1**

**Rating:** 6

**Review:**

1. Summary

The authors learn structured communication patterns between multiple RL agents. Their framework uses a Structured Communication Network Module and Communication-based Policy Module. These use a hierarchical decomposition of the multi-agent system and a graph neural network that operates over the resulting abstract agent (groups). The authors evaluate on two environments, where this approach outperforms other ways to communication protocols.

2. Decision (accept or reject) with one or two key reasons for this choice.

Weak accept.

3. Supporting arguments

Scalable communication with many agents will require a (learned) trade-off between structural priors and learning representations of communicaton (protocols). This work seems like an interesting step in analyzing how to implement this.

4. Additional feedback with the aim to improve the paper. Make it clear that these points are here to help, and not necessarily part of your decision assessment.

5. Questions

**Experience Assessment:**

I have published one or two papers in this area.

**Review Assessment: Checking Correctness Of Derivations And Theory:**

N/A

**Review Assessment: Checking Correctness Of Experiments:**

I assessed the sensibility of the experiments.

**Review Assessment: Thoroughness In Paper Reading:**

I made a quick assessment of this paper.

---

> ### Author Response · Authors · 2019-11-12
> **Answer for review #1**
>
> Thank you for your valuable and inspiring comments.
>
> The focus of the paper is to address the learning to communication problem in MARL. There are two key issues: communication topology and message representation. However, for large scale scenes, communication topology  (fully-connected, star, tree) in existing MARLs are not feasible for large scale scenes, and their message extraction methods (compute mean or LSTM of all agents' current state) are not strongly expressive, violate physical meanings, ignore the interaction between agents, or don't utilize the communication topology. Therefore, our motivation is to develop a flexible and scalable learning to communication approach for large-scale MARLs.
>
> We solve the two issue above by introducing: 1) a hierarchical structure communication, which dynamically learns the communication topology based on cluster-based-routing protocol; 2) the global message representation is deeply integrated in the local information aggregating and permeating through the learned structured network via graph neural network (GNN). The united learning of communication topology and graph-based message is first proposed in MARLs.

---

### Official Review · AnonReviewer2 · 2019-10-26
**Official Blind Review #2**

**Rating:** 3

**Review:**

Summary:
The paper proposes a method for improving the scalability of communication-based cooperative multi-agent reinforcement learning. While existing approaches assume a fixed underlying network topology over which agents communicate, in the proposed method, this network topology is dynamic (changes at each time step) and learnable (by assigning a weight to each node and "rewiring" nodes in a particular way based on these weights).

Authors highlight the importance of having a topology that is roughly similar to a collection of star-topologies. The center of stars (central nodes) further form a complete graph. They argue that such a topology can achieve global cooperation while reducing the number of messages exchanged as compared to the case where all agents can communicate with each other.

To learn a dynamically changing topology, the method assigns a weight (an integer between 0 and 4) to each agent based on its local observation. An existing method (CBRP) is then used to establish connections between agents based on weights assigned to them.

A graph neural network (GNN) is used for computing the messages that are exchanged among agents. Communication uses the following 3 steps: (i) agents talk to the central agent(s) to which they are connected, (ii) central agents exchange information among themselves, and (iii) central agents transmit information to the agents that are connected to them.

Each agent uses a deep Q-network - the parameters of this network are shared across agents. This Q-network receives rewards from the environment. Gradients flowing through Q-network are also used to update GNN. Since CBRP is non-differentiable, the parameters for network that computes weights for all the agents are also updated using a Q-network that gets the same reward from the environment as the first Q-network.

Experiments done on MAgent environment demonstrate that: (i) communication is useful and (ii) method scales well as number of agents increases. Additional qualitative studies have also been performed to understand the content of messages and the learned strategies. Authors have also experimented with the StarCraftII environment.


Comments:
The paper deals with an interesting problem, however, the presentation can be significantly improved as there are multiple grammatical mistakes in the manuscript. Unfortunately, the work does not position very well with the existing literature. The motivation and the impact of the contributions are not very clear. I would rate contributions as marginal.

It is not clear what POSs* terms in Algorithm 1 mean.

It would be interesting to see which agents become central agents over time. As central agents form a complete graph, if there are many central agents then the approach will be inefficient.

Under the message visualization heading on p9, it is not clear how one decides whether a message was a "move" message or an "attack" message.


Questions to the Authors:

1. On p2, it is written that when concatenation or mean operation is used for aggregation, then inter-relationship between agents are not captured. What does this mean? Why does this problem not apply to GNN based solution which may also use mean for aggregation?

2. On p10, first line, it is written that all methods were made to react to the same initial state. How was this state chosen?

A few questions are also embedded in the comments above.


**Experience Assessment:**

I have published one or two papers in this area.

**Review Assessment: Checking Correctness Of Derivations And Theory:**

N/A

**Review Assessment: Checking Correctness Of Experiments:**

I assessed the sensibility of the experiments.

**Review Assessment: Thoroughness In Paper Reading:**

I read the paper thoroughly.

---

> ### Author Response · Authors · 2019-11-12
> **Answer for review #2**
>
> Thank you for your valuable and inspiring comments.
> [Novelty]
> The focus of the paper is to address the learning to communication problem in MARL. There are two key issues: communication topology and message representation. However, for large scale scenes, communication topology  (fully-connected, star, tree) in existing MARLs are not feasible for large scale scenes, and their message extraction methods (compute mean or LSTM of all agents' current state) are not strongly expressive, violate physical meanings, ignore the interaction between agents, or don't utilize the communication topology. Therefore, our motivation is to develop a flexible and scalable learning to communication approach for large-scale MARLs.
>
> Our contribution is the proposed learning-structured-communication framework. We solve the two issue above by introducing: 1) a hierarchical structure communication, which dynamically learns the communication topology based on cluster-based-routing protocol; 2) the global message representation is deeply integrated in the local information aggregating and permeating through the learned structured network via graph neural network (GNN). The united learning of communication topology and graph-based message is first proposed in MARLs.
>
> The direct impact of our approach is that the proposed LSC can deal with larger scale problem than existing MARLs. With limited GPU resource, our method achieved better reward and game performances in larger scale settings than referred papers.
>
> [Central agents over time]
> The number of central agents is determined by the CBRP algorithm in structured communication network module. CBRP has two advantages: 1) it can dynamically decide the number of central agents; 2) the central nodes is exclusive to each other. The latter one can prevent the issue you concerned happen, i.e., too many central agents.
>
> ["Visualization"]
> Figure 5(c) and 5(d) show the visualization of message distribution, where the 3D dots stands for message in the message space, and the color and label is the corresponding action ('move' and 'attack') of agents. The action is decided by the agent's policy after receiving the message. We also have polished the  note of the Figure 5(c) and 5(d) for more clear presentation.
>
> [Why can not capture the inter-relationship]
>  When aggregating agents' observations, the concatenation and mean operation simply construct the same vector or value for agents, therefore the pair-to-pair relationship between agents can not be expressed. For GNN-based message representation, it may also use mean for aggregation. But the mean operate is executed for every agent only in their neighbour (not all the nodes) in the network.  Thus, the inter-relationship can be learned via GNN with the graph structure, which is also the advantage of GNN.
>
> ["Initial state" on Page 10]
> The initial state is chosen from the inference procedure of the "IDQN" method. The IDQN method works very bad after this state occurs. As a result, we consider that this state is difficult for IDQN and need to be solved. Then the other methods are all introduced to perform inference on this state. The comparisons are made based on this initial state.
>
> [POSs* terms][Presentation]
> THe "POSs_i^t" term means the position of agent i at step t. We have polished the description of it in the paper. We have checked the whole paper and made the presentation more readable. Also we have corrected many grammar mistakes.

---

### Official Review · AnonReviewer3 · 2019-10-31
**Official Blind Review #3**

**Rating:** 3

**Review:**

This paper proposes a method of learning a hierarchical communication graph for improving collaborative multi-agent reinforcement learning, particularly with large numbers of agents. The method is compared to a suitable range of baseline approaches across two complex environments. The initial results presented seem promising, but further work is needed to ensure the results are reproducible and repeatable.

To enable reproducibility, please include details of all hyperparameters used for all approaches in both domains. These should include justification of how the hyperparameters were tuned. Without understanding how these values were set I cannot support acceptance.

To ensure the results are repeatable, repeated runs of training should be completed and the variation in performance quantified in the results. These repeats may have already been performed as Table 3 and Figure 5 discuss average results, but if not they must be completed before the work can be published due to the known issues with high variance in performance that commonly occur in deep RL.

I would also argue against the justification of excluding ATOC from the StarCraft II experiments as its performance in MAgent with 25 agents is comparable to the other baseline methods that were tested. However, this is lower priority than the issues above provided there is no significant change in the relative performance of methods when the variance across multiple runs is documented in all existing experiments.

Minor Comments:
The following are suggestions for improvements if the paper is accepted or for future submissions.

In Section 2, centralised critic methods are grouped as "communication-free" however I don't think this is the best term to explain this approach as each agent has to communicate both its observations and actions to a centralised node (e.g. COMA) or all other agents (e.g. MADDPG). I also think this section should include coverage of other methods of utilizing graph neural networks in multi-agent reinforcement learning - e.g. "Deep Multi-Agent Reinforcement Learning with Relevance Graphs." Malysheva et al. Deep RL Workshop @ NeurIPS 2018 and "Relational Forward Models for Multi-Agent Learning" Tacchetti et al. ICLR 2019.

In Section 3, the acronyms CBRP and HCOMM are used on page 4 before they are introduced in full on page 5 for CBRP and never for HCOMM. HCOMM is also not used in Figure 4 or in the text description of the method. I believe it is the module described in Section 3.2 but this should be made clearer.

Many claims in the paper are worded too strongly and should be revised. In Section 2, it is claimed that DQN "is one of the few RL algorithms applicable for large-scale MARL" - However, there are now many successful applications of deep RL to multi-agent systems (some of which are cited earlier in this same section) that use a variety of algorithms other than DQN. It is also claimed in this section that "DQN has excellent sample efficiency" despite the sample efficiency of deep RL being a known issue, open research question and a barrier to its widespread use in practice.

In Section 3.2 the authors conclude "Overall, our LSC algorithm has advantage in the communication efficiency" despite in the same section noting two cases where ATOC has better efficiency (N_msg and N_b-r). I would suggest removing this sentence entirely as the paragraph above already contains a balanced account of the relative merits of each approach.

On pages 8 and 9 the authors make references to guarantees and in the Appendix to proofs that are not supported by theory only empirical results. Without supporting theory these words should be avoided.

The writing is also often informal to the detriment of presenting important information clearly. Notably, on page 3 "due to explosive growing number of agents" and on page 4 "The overall task of the MARL problem can be solved by properly objective function modeling." The second of these, particularly the word "solved" is also related to the issue above of using the words guarantee and prove.

Finally, the paper would benefit from a thorough grammar check. I note the following issues as simple changes that can be made to improve the readability of the paper:
- In the abstract, the sentence "but also communication high-qualitative" does not parse. Perhaps this could be shortened to the brackets following? i.e. "is not only scalable but also learns efficiently."
- Page 2, "or employing the LSTM to" -> "or use the LSTM to"
- Page 2, "still hinder the" -> "still hinders the"
- Page 2, "the communication structure need be jointly" -> needs to be jointly
- Page 6, "policy module motioned below" -> discussed below
- Page 10, "the map size are is 1920x1200" -> the map size is 1920x1200
- Page 10, "We do not compare with ATOC because its poor performance in MAgent" -> because of its
- Page 10, "we will to improve" -> we will improve
- Page 10, "practical constrains" -> practical constraints

**Experience Assessment:**

I have published one or two papers in this area.

**Review Assessment: Checking Correctness Of Derivations And Theory:**

I assessed the sensibility of the derivations and theory.

**Review Assessment: Checking Correctness Of Experiments:**

I carefully checked the experiments.

**Review Assessment: Thoroughness In Paper Reading:**

I read the paper thoroughly.

---

> ### Author Response · Authors · 2019-11-12
> **Answer for review #3**
>
> Thank you for your valuable and inspiring comments.
> [To enable reproducibility] [About all hyperparameters]
> We provide the details of hyperparameters used for all approaches in both domains in the appendix, see section A.2. For MAgent, all approaches use the same hyperparameters,  except that ATOC can be hard to learn with the same hyperparameters settings, thus we tuned the message dimension to 64 for ATOC. For the radius of communication, IDQN, CommNet and IC3 have specific requirements, thus we keep other algorithms using same radius of communication. For the network design, we choose similar parameters, similar embedding and aggregation layers except ATOC. ATOC [https://arxiv.org/abs/1805.07733] specify the Recurrent aggregation, thus we use a GRU cell to aggregate messages in ATOC while others use same segment sum operator to do aggregation. For SC2, all approaches share similar parameters as in smac [https://arxiv.org/abs/1902.04043]. For network design, we use a similar design like in MAgent, only tuned layers to adapt different input and out. For more details, refer to section A.2.
>
> [To ensure repeatable]
> We have completed the repeated runs of training for MAgent, and update the results in the paper (see Figure 5a). As shown in the new Figure 5a, the averaged training reward and the reward variance of LSC, with 5 repeated runs, are outperform other approaches when algorithm converged.
> For Starcraft2, we are repeatedly running the algorithms following your advice. The deep MARLs training in large scale and complex Starcraft2 cost many computation and GPU resources, but unfortunately our resource is limited. We will update Figure 7 with variations before the rebuttal deadline or as soon as the programs finish. However, from the current Figure 7, it can be noted that the performance superiority of our LSC to other methods is even bigger than in MAgent.
>
> [Excluding ATOC from the SC II experiment]
> We have already run ATOC in StarCraft2 environment, but the results are not good enough to present. We are repeated running all the compared algorithms with different random seeds, and at the same time we are trying our best to tune ATOC to obtain better performance. The result of ATOC on SC2 will be updated once the ongoing experiments finish, (hopefully) before rebuttal deadline, as well as the repeated training results of all compared algorithms.
>
> [Minor Comments] Thanks for your suggestions. We have checked and polished the whole paper to make the presentation more accurate and clear.

---

> > ### Author Response · Authors · 2019-11-15
> > **Paper update**
> >
> > We update the paper again.
> >
> > The repeated training results of all compared algorithms including ATOC are updated in the new Figure 7, which demonstrates the advantage of the proposed LSC algorithm in SCII enviroment.

---

### Decision · Program_Chairs · 2019-12-19

**Decision:**

Reject

**Comment:**

The paper focuses on large-scale multi-agent reinforcement learning and proposes Learning Structured Communication (LSC) to deal issues of scale and learn sample efficiently. Reviewers are positive about the presented ideas, but note remaining limitations. In particular, the empirical validation does not lead to sufficiently novel insights, and additional analysis is needed to round out the paper.